# ABO blood groups are not associated to gestational diabetes mellitus in Mexican women

Hid Felizardo Cordero-Franco[1]*, Ana María Salinas-Martínez[1,2], María José Esparza-Contró[3,4], Sofía Denisse González-Rueda[1], Francisco Javier Guzmán-de la Garza[1,5]

1 Unidad de Investigación Epidemiológica y en Servicios de Salud/CIBIN, Instituto Mexicano del Seguro Social, Monterrey, Nuevo León, México, 2 Facultad de Salud Pública y Nutrición, Universidad Autónoma de Nuevo León, Monterrey, Nuevo León, México, 3 Vicerrectoría de Ciencias de la Salud, Universidad de Monterrey, Monterrey, Nuevo León, México, 4 Unidad de Medicina Familiar No. 26, Instituto Mexicano del Seguro Social, Monterrey, Nuevo León, México, 5 Facultad de Medicina, Universidad Autónoma de Nuevo León, Monterrey, Nuevo León México

* hid.cordero@imss.gob.mx, dr_hid_cordero@hotmail.com

## Abstract

### Objectives

Some studies show an increased risk of gestational diabetes mellitus for ABO blood groups. Others find a lower risk or do not identify any association. Inconsistencies may be due to the heterogeneity in the control for confounding variables. We determined the association between ABO blood groups and gestational diabetes mellitus in Mexican women, controlling for gravidity and age, pre-pregnancy body mass index, fasting glucose at the first trimester, and first-degree relative with diabetes.

### Methods

This case-control study was conducted from February 2019 to December 2021 in Monterrey, Mexico, with 185 cases (women with gestational diabetes mellitus) and 530 controls. ABO blood groups and other variables were obtained from the clinical records. A multivariate binary logistic regression was used for estimating association. Two models were run, one for primigravidae and another for non-primigravidae. A p-value < 0.05 was significant.

### Results

The ABO blood groups were O (69.4%), A (22.2%), B (6.7%), and AB (1.7%), with no differences between cases and controls (p = 0.884). No association was found between ABO blood groups and gestational diabetes mellitus, in primigravidae or non-primigravidae.

### Conclusion

ABO blood groups were not associated with an increased risk of gestational diabetes mellitus in Mexican women, independent of gravidity and well-known risk factors.

**Data Availability Statement:** All relevant data are within the paper and its Supporting information files.

**Funding:** The author(s) received no specific funding for this work.

**Competing interests:** The authors have declared that no competing interests exist.

## Introduction

Gestational diabetes mellitus (GDM) is a condition characterized by varying degrees of carbohydrate intolerance that typically arises during pregnancy [1]. Its prevalence has been on the rise in parallel with the increasing rates of obesity and diabetes in the general population [2, 3]. Globally, the reported prevalence stands at 14%, with figures ranging from 6% to 24% in the United States and 13% in Mexico [4–6]. Several risk factors contribute to the development of GDM, including a family history of diabetes, advanced maternal age (older than 30), history of abortions, unexplained fetal deaths, and a previous diagnosis of GDM [3, 7]. Timely diagnosis is crucial as it significantly reduces the risk of maternal complications such as polyhydramnios and preeclampsia, as well as neonatal complications like preterm birth, respiratory distress, macrosomia, birth injury, and hypoglycemia [1].

The ABO system antigens are complex carbohydrates found on the surface of erythrocytes and various tissues; each antigen follows an autosomal dominant Mendelian inheritance pattern [8]. The ABO blood type gene is located on chromosome 9 (9q34.1) and is known as ABO glycosyltransferase [9]. There are three primary allelic forms: A and B, which are codominant, and O, which is recessive. The A and B alleles encode slightly different glycosyltransferases that add N-acetyl galactosamine and D-galactose, respectively, to substance H, a precursor side chain that is ultimately transformed into either the A or B antigen. However, the O allele does not produce a functional enzyme due to a mutation [10].

In addition to their significance in transfusion medicine, ABO blood groups have been associated with a predisposition to various infectious, neoplastic, and cardiovascular diseases [10], including type 2 diabetes (this one more consistently with the B blood type) [11–14]. The biological mechanisms underlying the association between ABO groups and diabetes remain unclear. For instance, tumor necrosis factor-α, a pro-inflammatory protein, can disrupt insulin receptor signaling and β-cell function, leading to hyperglycemia. Additionally, soluble intracellular cell adhesion molecule-1 (sICAM-1) on the surface of endothelial cells impairs endothelial function, which is also compromised in women with GDM [2]. Genetic variations of the ABO groups have been linked to tumor necrosis factor-α, sICAM-1, and selectin (another marker of endothelial dysfunction related to diabetes mellitus) [15–17]. However, epidemiological evidence on the association between ABO blood groups and GDM is inconsistent. Some studies indicate a higher risk for the A group [18, 19], the B group [19], the O group [19, 20], and the AB group [21–24]. Conversely, others report a lower risk for the AB group [19, 25] or fail to identify any association [26–29]. These discrepancies may arise from variations in controlling for confounding variables, with gravidity being an important factor to consider [30–32]. Primigravidae and non-primigravidae differ in their exposure to risk factors such as a history of GDM, macrosomia, and stillbirths or spontaneous abortions. Therefore, it is crucial to analyze the association between ABO blood groups and GDM separately for primigravidae and non-primigravidae. Surprisingly, to date, no studies have explored this association while stratifying by gravidity status.

The objective of this study was to investigate the association between ABO blood groups and GDM in Mexican women, while stratifying for the number of pregnancies (primigravidae and non-primigravidae) and adjusting for other well-established risk factors, including age, pre-pregnancy body mass index, fasting glucose levels in the first trimester, and a first-degree relative with diabetes.

## Materials and methods

This case-control study was conducted at a primary care center in the metropolitan area of Monterrey, Mexico, from February 2019 to December 2021. The cases consisted of patients

diagnosed with (GDM) between the 24th and 28th week of pregnancy. GDM diagnosis was established following the American Diabetes Association guidelines available at the time of the study, using the one-step strategy which involved performing an oral glucose tolerance test-75 g in the morning after an overnight fast of at least 8 hours. GDM diagnosis was made if any of the following values were met or exceeded: 92 mg/dL fasting, 180 mg/dL one hour after the oral glucose, and 153 mg/dL two hours after the oral glucose [33]. Each case was matched with more than two controls (women who did not meet the diagnostic criteria for GDM). To be included in the study, all participants had to have initiated prenatal care before the 14th week of pregnancy. Women with prediabetes, diabetes, using metformin for polycystic ovaries, or on glucocorticoids were excluded. Cases and controls were consecutively included after verifying the selection criteria. Initially, a sample size of 166 cases and 332 controls was estimated based on an expected odds ratio of 2.7, with 19.5% of cases and 9.8% of controls exposed to blood group AB [21], a statistical power of 80%, and alpha of 5% [34]. However, the final sample consisted of 185 cases and 530 controls, with no missing data.

## Ethical considerations

The research protocol was approved by the Local Committee of Ethics and Health Research (R-2019-1912-016). Written informed consent was obtained from all participants before their inclusion, and the study adhered to the guidelines of the Declaration of Helsinki for Research on Human Subjects [35].

## Variables

The predictor variable was ABO blood group, and the outcome variable was GDM. Data on the ABO blood group, Rh factor, pre-pregnancy body mass index, weight gain during pregnancy (from the date of the last menstrual period to the date of GDM screening), first fasting glucose level of the first trimester, obstetric history (including the number of pregnancies [gravidity], stillbirths, spontaneous abortions, previous preeclampsia or GDM, and macrosomia), medical history (such as pre-pregnancy chronic hypertension, polycystic ovarian syndrome, and having a first-degree relative with diabetes), and sociodemographic profile (including age, marital status, occupation, and education) were collected from the clinical records. These data were then verified through face-to-face interviews with the patients, following a standard clinical history format. The interviews were conducted by two medical residents who had undergone prior training and received continuous supervision. The interviews took place in a private office located near the clinical laboratory.

Blood glucose levels were analyzed using the photometry method with the COBAS C 501 equipment from Roche (Germany). The determination of the ABO blood groups, and Rh factor was performed in the clinical laboratory upon the request of the treating physician at the beginning of pregnancy. This involved conducting a search for hemagglutination reactions by placing capillary blood from the patients on different slides and mixing it with anti-A, anti-B, anti-AB, and anti-D sera. Agglutination was observed according to blood type.

## Statistical analysis

Measures of central tendency and dispersion were estimated for numerical variables, while proportions were calculated for categorical variables. The Mann-Whitney U test was used to compare numerical variables between cases and controls after verifying that they did not follow a normal distribution using the Kolmogorov-Smirnov test. The Chi-square test and binary logistic regression were employed to analyze the association between ABO blood groups and other categorical and numerical variables, respectively, with GDM. A multivariate binary

logistic regression was used to estimate odds ratios (OR) and 95% confidence intervals (CI), with the ABO blood group serving as the predictor variable (with the O group as the reference) and GDM status as the outcome variable. Separate models were run for women in their first pregnancy and women with two or more pregnancies. The control variables in both models included age, pre-pregnancy body mass index, first-trimester fasting glucose, a first-degree relative with diabetes, weight gain at the GDM screening date, and polycystic ovary syndrome. Additional variables such as the history of spontaneous abortions, GDM, and macrosomia were included in the non-primigravidae model. A p-value < 0.05 was considered statistically significant.

## Results

The study included a total of 185 cases and 530 controls. Of the total sample, 473 women had two or more pregnancies (66.2%). Cases were significantly older than controls (p < 0.0001), had a higher pre-pregnancy body mass index (p < 0.0001), and exhibited higher levels of first fasting glucose in the first trimester (p < 0.0001). Furthermore, cases showed a higher frequency of pre-pregnancy chronic hypertension (p < 0.05), a history of two or more pregnancies (p < 0.05), previous GDM (p < 0.0001), and macrosomia (p < 0.05). The most common blood group among the participants was O (69.4%), followed by A (22.2%), B (6.7%), and AB (1.7%). However, there were no significant differences in the distribution of blood groups between cases and controls (p = 0.561). The Rh factor positive was the most prevalent (97.2%), with no significant difference between women with and without GDM (p = 0.436; Table 1).

### Association between ABO blood groups and GDM

Among women in their first pregnancy (n = 242), no significant association was found between ABO blood groups and GDM. Only age ≥ 30 and first fasting glucose levels in the first trimester were associated with an increased risk of GDM (Table 2).

Similarly, in women with two or more pregnancies (n = 473), no significant association was observed between ABO blood groups and GDM. Age ≥ 30, first trimester glucose levels, and previous GDM were the predominant risk factors in this group (Table 3).

## Discussion

This study aimed to investigate the association between ABO blood groups and GDM in Mexican women, considering gravidity and controlling for other established risk factors. Before discussing the main findings, it is crucial to highlight the representativeness of the sample studied. The participants were pregnant women attending an urban primary care center, predominantly aged under 30, and a significant proportion had experienced two or more pregnancies. This profile aligns with the target population and enhances the generalizability of our findings.

As anticipated, women with GDM displayed a higher frequency of known risk factors compared to those without GDM [3, 7]. Additionally, they exhibited higher levels of first-trimester fasting glycemia. It is noteworthy that elevated first-trimester fasting glycemia has recently been recognized as a prognostic factor for GDM, even at levels considered within the normal range for individuals without diabetes, starting at 4.735 mmol/L (85 mg/dL) [36]. These clinical observations underscore the significance and relevance of investigating the potential association between ABO blood groups and GDM within this specific sample. Consequently, it was crucial to meticulously control for the effects of these variables in the association analysis.

In this study, we did not find an association between ABO blood groups and GDM. This finding could be attributed to the rigorous statistical control of the confounding variables as

**Table 1. Sociodemographic, medical, and obstetric variables, and blood groups according to GDM status.**

| Variable | Cases n = 185 | Controls n = 530 | p-value* |
|---|---|---|---|
| *Sociodemographic* | | | |
| Age (years; mean ± SD) | 28.4 ± 5.7 * | 25.8 ± 5.4 | < 0.0001 |
| Marital status, living with her partner (%) | 154 (83.2) | 443 (83.6) | 0.909 |
| Occupation, economically active (%) | 115 (62.2) | 295 (55.7) | 0.142 |
| Education, high school, and higher (%) | 118 (63.8) | 349 (65.8) | 0.654 |
| *Current pregnancy* | | | |
| Pre-pregnancy body mass index (kg/m$^2$; mean ± SD) | 28.3 ± 6.7 * | 26.0 ± 5.4 | < 0.0001 |
| Weight gain at GDM screening date (kg; mean ± SD) | 5.7 ± 5.1 | 6.3 ± 5.2 | 0.220 |
| First fasting glucose in the first trimester (mg/dL; mean ± SD) | 88.3 ± 9.6 * | 82.3 ± 7.0 | < 0.0001 |
| *Medical history* | | | |
| Pre-pregnancy chronic hypertension (%) | 6 (3.2) * | 4 (0.8) | 0.023 |
| Polycystic ovary syndrome (%) | 6 (3.2) | 23 (4.3) | 0.666 |
| First-degree relative with diabetes (%) | 71 (38.4) | 165 (31.1) | 0.084 |
| *Obstetric history* | | | |
| Two and more pregnancies (%) | 140 (75.7)* | 333 (62.8) | 0.002 |
| Stillbirths or spontaneous abortions (%) [a] | 43 (30.7) | 103 (30.9) | 1.000 |
| Previous preeclampsia (%) [a] | 12 (8.6) | 21 (6.3) | 0.237 |
| Previous GDM (%) [a] | 18 (12.9) * | 12 (3.6) | < 0.0001 |
| Macrosomia (%) [a] | 19 (13.7) * | 23 (6.9) | 0.032 |
| *ABO blood groups* | | | 0.884 |
| O group (%) | 127 (68.6) | 369 (69.6) | |
| A group (%) | 40 (21.6) | 119 (22.5) | |
| B group (%) | 14 (7.6) | 34 (6.4) | |
| AB group (%) | 4 (2.2) | 8 (1.5) | |
| *Rh factor* | | | |
| Rh positive | 178 (96.2) | 517 (97.5) | 0.436 |

[a] Only women with ≥ 2 pregnancies.

GDM: Gestational diabetes mellitus. SD: Standard deviation.

* Mann-Whitney U test or Chi-square test, as appropriate.

well as the stratification by gravidity. However, the results of other studies in this field are highly contradictory. Phaloprakarn et al. [26] and Sajan et al. [28] did not find an association after controlling for gravidity and GDM risk factors. Conversely, Zhang et al. [19] identified an increased risk with groups A, B, and O, in a study with similar adjustments. Huidobro et al. [18] and Sapanont et al. [20] found associations with groups A and O, respectively, after controlling for risk factors, but not gravidity. Studies reporting the AB blood group present even more conflicting results: Shimodaira et al. [21] reported a 2.73-fold increased risk of developing GDM in women with group AB, while Rom et al. [25] identified a decreased risk, both studies after adjusting for risk factors and gravidity. Other authors [22–24] found no association but did not perform a statistical adjustment. It is important to consider gravidity in the analysis since a woman who has experienced a previous pregnancy might have an increased risk due to a history of GDM, macrosomia, stillbirths, or spontaneous abortions.

The biological evidence behind the association between ABO blood groups and GDM at a pathophysiological level is still incipient. High levels of molecules important in inflammatory processes, such as sP-selectin, sICAM-1 (both cell adhesion molecules), and tumor necrosis

**Table 2. Association between the ABO blood groups, other well-known risk factors and GDM in women in their first pregnancy.**

| Variables | OR crude (95% CI) | OR adjusted (95% CI) |
|---|---|---|
| Blood group | | |
| O (reference) | 1.0 | 1.0 |
| A | 1.0 (0.5, 2.2) | 1.1 (0.5, 2.5) |
| B | 0.4 (0.0, 2.8) | 0.4 (0.0, 3.2) |
| AB | 1.1 (0.1, 9.8) | 1.2 (0.1, 15.0) |
| Age $\geq$ 30 | 2.8 (1.3, 6.0) * | 2.5 (1.1, 5.8) * |
| Pre-pregnancy body mass index 25–29.9 kg/m$^2$ | 1.3 (0.6, 2.8) | 1.0 (0.4, 2.2) |
| Pre-pregnancy body mass index $\geq$ 30 kg/m$^2$ | 1.7 (0.7, 3.7) | 1.1 (0.4, 2.6) |
| First fasting glucose in the first trimester [a] | 1.1 (1.0, 1.2) * | 1.1 (1.0, 1.2) * |
| First-degree relative with diabetes | 1.2 (0.6, 2.3) | 1.0 (0.4, 2.2) |
| Weight gain at GDM screening date [b] | 1.0 (0.9, 1.1) | 1.0 (0.9, 1.1) |
| Polycystic ovary syndrome | 1.8 (0.6, 5.8) | 1.5 (0.4, 5.6) |

[a] Increased risk for each mg/dL of blood glucose.

[b] Increased risk for each kilogram gained.

GDM: Gestational diabetes mellitus. CI: Confidence interval.

* p- value < 0.05.

factor alpha (TNF-α, a proinflammatory cytokine), have been linked to GDM [2]. Also, the A blood group has been associated with an increase in the circulating levels of sP-selectin and sICAM-1 [15, 17], while a genome-wide analysis revealed higher circulating levels of TNF-α in people with group O [16]. Further research is needed to elucidate the reasons for these

**Table 3. Association between the ABO blood groups, other well-known risk factors and GDM in women with two and more pregnancies.**

| Variables | OR crude (95% CI) | OR adjusted (95% CI) |
|---|---|---|
| Blood group | | |
| O (reference) | 1.0 | 1.0 |
| A | 1.0 (0.6, 1.6) | 1.0 (0.6, 1.7) |
| B | 1.4 (0.7, 3.0) | 1.3 (0.6, 2.9) |
| AB | 1.8 (0.4, 8.4) | 1.4 (0.3, 7.6) |
| Age $\geq$ 30 | 2.2 (1.4, 3.3) * | 1.8 (1.1, 2.9) * |
| Pre-pregnancy body mass index 25–29.9 kg/m$^2$ | 1.3 (0.8, 2.1) | 0.9 (0.6, 1.6) |
| Pre-pregnancy body mass index $\geq$ 30 kg/m$^2$ | 1.9 (1.2, 3.2) * | 1.2 (0.7, 2.1) |
| First fasting glucose in the first trimester [a] | 1.1 (1.0, 1.2) * | 1.1 (1.0, 1.2) * |
| First-degree relative with diabetes | 1.4 (0.9, 2.1) | 1.1 (0.7, 1.8) |
| Stillbirths or spontaneous abortions | 1.0 (0.6, 1.5) | 1.0 (0.6, 1.5) |
| Previous GDM | 3.9 (1.9, 8.3) * | 3.3 (1.4, 7.6) * |
| Macrosomia | 2.1 (1.1, 4.0) * | 1.5 (0.7, 3.1) |
| Weight gain at GDM screening date [b] | 1.0 (0.9, 1.1) | 1.0 (0.9, 1.0) |
| Polycystic ovary syndrome | 0.4 (0.1, 1.4) | 3.1 (0.6, 15.9) |

[a] Increased risk for each mg/dL of blood glucose.

[b] Increased risk for each kilogram gained.

GDM: Gestational diabetes mellitus. CI: Confidence interval.

*p-value < 0.05.

inconsistencies, beyond the number of pregnancies, in order to make recommendations regarding the role of blood groups in the development of GDM.

This study has some limitations. It would have been desirable to have a larger sample size, especially for the less frequent ABO groups, such as B and AB. However, the distribution of blood groups observed in our study was very similar to that reported in Mexico [37]. One strength of this study was the rigorous control of confounding variables, particularly gravidity, along with age, pre-pregnancy body mass index, fasting glucose in the first trimester, polycystic ovary syndrome, weight gain before the diagnosis of GDM, and the presence of a first-degree relative with diabetes.

In conclusion, our findings indicate that ABO blood groups were not associated with an increased risk of GDM in Mexican women, independent of the number of pregnancies and other well-known risk factors. Studies such as this, with rigorous statistical analysis, contribute to the body of knowledge concerning the role of blood groups in the development of GDM. Further investigations are warranted to gain a better understanding of the inconsistencies observed in the literature and to provide more definitive recommendations regarding the relationship between blood groups and the risk of developing GDM.

## Supporting information

**S1 Checklist. STROBE statement—Checklist of items that should be included in reports of *case-control studies.***
(DOCX)

## Acknowledgments

The authors wish to thank the valuable help of the clinical laboratory staff of the Family Medicine Unit No. 26 of the Mexican Institute of Social Security.

## Author Contributions

**Conceptualization:** Hid Felizardo Cordero-Franco, Ana María Salinas-Martínez, María José Esparza-Contró.

**Data curation:** Hid Felizardo Cordero-Franco, Ana María Salinas-Martínez, María José Esparza-Contró, Sofía Denisse González-Rueda, Francisco Javier Guzmán-de la Garza.

**Formal analysis:** Hid Felizardo Cordero-Franco, Sofía Denisse González-Rueda, Francisco Javier Guzmán-de la Garza.

**Investigation:** Hid Felizardo Cordero-Franco, María José Esparza-Contró, Sofía Denisse González-Rueda, Francisco Javier Guzmán-de la Garza.

**Methodology:** Hid Felizardo Cordero-Franco, Ana María Salinas-Martínez, María José Esparza-Contró.

**Project administration:** Hid Felizardo Cordero-Franco, María José Esparza-Contró, Sofía Denisse González-Rueda, Francisco Javier Guzmán-de la Garza.

**Software:** Hid Felizardo Cordero-Franco, Sofía Denisse González-Rueda.

**Supervision:** Hid Felizardo Cordero-Franco, Ana María Salinas-Martínez, Sofía Denisse González-Rueda, Francisco Javier Guzmán-de la Garza.

**Validation:** Hid Felizardo Cordero-Franco, Ana María Salinas-Martínez, Sofía Denisse González-Rueda, Francisco Javier Guzmán-de la Garza.

**Writing – original draft:** Hid Felizardo Cordero-Franco, Ana María Salinas-Martínez, María José Esparza-Contró.

**Writing – review & editing:** Hid Felizardo Cordero-Franco, Ana María Salinas-Martínez, María José Esparza-Contró, Sofía Denisse González-Rueda, Francisco Javier Guzmán-de la Garza.

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
