## [Decision Letter · Decision Letter 0]

30 Aug 2023

PONE-D-23-17533ABO blood groups are not associated to gestational diabetes mellitus in Mexican womenPLOS ONE

Dear Dr. Cordero-Franco,

Thank you for submitting your manuscript to PLOS ONE. After careful consideration, we feel that it has merit but does not fully meet PLOS ONE’s publication criteria as it currently stands. Therefore, we invite you to submit a revised version of the manuscript that addresses the points raised during the review process.

We look forward to receiving your revised manuscript.

Kind regards,

Frank T. Spradley

Academic Editor

PLOS ONE

Journal Requirements:

2. Please ensure that you include a title page within your main document. You should list all authors and all affiliations as per our author instructions and clearly indicate the corresponding author.

4. Please review your reference list to ensure that it is complete and correct. If you have cited papers that have been retracted, please include the rationale for doing so in the manuscript text, or remove these references and replace them with relevant current references. Any changes to the reference list should be mentioned in the rebuttal letter that accompanies your revised manuscript. If you need to cite a retracted article, indicate the article’s retracted status in the References list and also include a citation and full reference for the retraction.

Reviewers' comments:

Reviewer's Responses to Questions

**Comments to the Author**

1. Is the manuscript technically sound, and do the data support the conclusions?

Reviewer #1: Yes

Reviewer #2: Yes

2. Has the statistical analysis been performed appropriately and rigorously? 

Reviewer #1: Yes

Reviewer #2: Yes

3. Have the authors made all data underlying the findings in their manuscript fully available?

Reviewer #1: Yes

Reviewer #2: Yes

4. Is the manuscript presented in an intelligible fashion and written in standard English?

Reviewer #1: Yes

Reviewer #2: Yes

5. Review Comments to the Author

Reviewer #1: An interesting article similar to others proposed in the background of the text, but with negative results. It is an interesting "me too" work with potential use in systematic reviews, with a clearly formulated research question that uses explicit and systematic methods to identify, select and critically evaluate the relationship studied in Latin America.

Reviewer #2: COMMENTS FOR THE AUTHOR:

This case-control study examined associations between ABO blood groups and GDM in Mexican women, controlling for several well-known factors. The results found ABO blood groups were not associated with an increased risk of GDM in Mexican women. This manuscript would benefit greatly from a novel perspective and careful writing.

Specific comments:

Line 43. I noticed that ABO blood groups have been associated with a predisposition to type 2 diabetes, but the authors did not mention which blood group affect the risk of type 2 diabetes. And was the evidence consistent for type 2 diabetes? Please clarity the background.

Line 56-59. Please add references regarding the gravidity being an important factor.

In the introduction section (line 28-30), the authors mentioned a series of factors contributing to the development of GDM. Also, plenty of variables have been collected in the study (line 92-108). However, most of the factors were not considered in the current analysis. Did it affect the results? Additionally, weight gain before diagnosis of GDM has been identified as a risk factor for GDM in other studies, should it be adjusted in the model?

The variable “gravidity” was similar to “parity”. Usually, the number of gravidity was not the same to that of parity. Why do the authors choose gravidity instead of parity in the study?

Line 138. Line 170. The full title of GDM should not be written here. Please also check the expression through the paper.

In the results section, age >=30 was found to be associated with GDM. Is it possible to conduct a stratified analyses by age to explore the association?

Please add more explanation of mechanisms in the discussion section.

6. PLOS authors have the option to publish the peer review history of their article (what does this mean?). If published, this will include your full peer review and any attached files.

Reviewer #1: No

Reviewer #2: **Yes: **Xi Chen

---

## [Author Response · Author response to Decision Letter 0]

14 Sep 2023

Dear reviewers:

Thank you for your valuable comments. The authors have addressed all observations and, if applicable, made the suggested corrections to improve the quality of the manuscript. Changes to the full text appear in the “Revised Manuscript with Track Changes”.

Review Comments to the Author

Reviewer #1: An interesting article similar to others proposed in the background of the text, but with negative results. It is an interesting "me too" work with potential use in systematic reviews, with a clearly formulated research question that uses explicit and systematic methods to identify, select and critically evaluate the relationship studied in Latin America.

Reviewer #2: COMMENTS FOR THE AUTHOR:

This case-control study examined associations between ABO blood groups and GDM in Mexican women, controlling for several well-known factors. The results found ABO blood groups were not associated with an increased risk of GDM in Mexican women. This manuscript would benefit greatly from a novel perspective and careful writing.

Specific comments:

1. Line 43. I noticed that ABO blood groups have been associated with a predisposition to type 2 diabetes, but the authors did not mention which blood group affect the risk of type 2 diabetes. And was the evidence consistent for type 2 diabetes? Please clarity the background. Answer: the B blood type was more consistently associated with type 2 diabetes. More references about it were added, and that mention was added in the background section.

2. Line 56-59. Please add references regarding the gravidity being an important factor. Answer: done. References were added to that assertion.

3. In the introduction section (line 28-30), the authors mentioned a series of factors contributing to the development of GDM. Also, plenty of variables have been collected in the study (line 92-108). However, most of the factors were not considered in the current analysis. Did it affect the results? Answer: thanks for this observation. Almost all the mentioned variables had been previously included in the models, except polycystic ovary syndrome and weight gain before diagnosis of GDM. Thanks to your valuable comment, these variables were added in both models (primigravidas and non-primigravidas), and their inclusion did not affect the results of the new version of the manuscript. The complete revised results are in Tables 2 and 3.

4. Additionally, weight gain before diagnosis of GDM has been identified as a risk factor for GDM in other studies, should it be adjusted in the model? Thanks for your comment. We added weight gain before GDM diagnosis to both models, and final results did not significantly change. The complete revised results are in Tables 2 and 3.

5. The variable “gravidity” was similar to “parity”. Usually, the number of gravidity was not the same to that of parity. Why do the authors choose gravidity instead of parity in the study? Answer: You are right: gravidity refers to the number of pregnancies, complete or incomplete, experienced by a female. It is different from PARITY, which is the number of offspring borne. We decided to analyze “gravidity” because separating a group of women in their first pregnancy (primigravidas) allowed us to have a group without abortions, stillbirths or even prior GDM. Thus, the search for the association between the ABO and DMG groups was more rigorous. Otherwise, if the group had been "primiparous", we could have unintentionally included women with those confounding factors, clouding a possible association.

6. Line 138. Line 170. The full title of GDM should not be written here. Please also check the expression through the paper. Answer: done. The full title was removed in those lines and others, using the abbreviation GDM through the paper, except in the abstract, according to the journal’s instructions for authors. 

7. In the results section, age >=30 was found to be associated with GDM. Is it possible to conduct a stratified analyses by age to explore the association? Answer: Thanks for your suggestion. We tried to stratify by age ≥ 30, along with gravidity, but in addition to not finding a statistically significant association between the variables under study, the sample sizes to compare were reduced much more, which, in our opinion, would lose precision in our estimates.

8. Please add more explanation of mechanisms in the discussion section. Answer: done. The explanation of mechanisms has been added.

---

## [Editor Report · Decision Letter 1]

21 Sep 2023

ABO blood groups are not associated to gestational diabetes mellitus in Mexican women

PONE-D-23-17533R1

Dear Dr. Cordero-Franco,

We’re pleased to inform you that your manuscript has been judged scientifically suitable for publication and will be formally accepted for publication once it meets all outstanding technical requirements.

Kind regards,

Frank T. Spradley

Academic Editor

PLOS ONE

---

## [Editor Report · Acceptance letter]

6 Oct 2023

PONE-D-23-17533R1 

ABO blood groups are not associated to gestational diabetes mellitus in Mexican women 

Dear Dr. Cordero-Franco:

I'm pleased to inform you that your manuscript has been deemed suitable for publication in PLOS ONE. Congratulations! Your manuscript is now with our production department. 

Kind regards, 

on behalf of

Dr. Frank T. Spradley 

Academic Editor

PLOS ONE